# Contraceptive progestins with androgenic properties stimulate breast epithelial cell proliferation

Marie Shamseddin[1,†,‡,§] [iD], Fabio De Martino[1,†] [iD], Céline Constantin[1] [iD], Valentina Scabia[1] [iD], Anne-Sophie Lancelot[1] [iD], Csaba Laszlo[1] [iD], Ayyakkannu Ayyannan[1] [iD], Laura Battista[1] [iD], Wassim Raffoul[2] [iD], Marie-Christine Gailloud-Matthieu[3] [iD], Philipp Bucher[1] [iD], Maryse Fiche[3] [iD], Giovanna Ambrosini[1] [iD], George Sflomos[1] [iD] & Cathrin Brisken[1,4,*] [iD]

## Abstract

Hormonal contraception exposes women to synthetic progesterone receptor (PR) agonists, progestins, and transiently increases breast cancer risk. How progesterone and progestins affect the breast epithelium is poorly understood because we lack adequate models to study this. We hypothesized that individual progestins differentially affect breast epithelial cell proliferation and hence breast cancer risk. Using mouse mammary tissue *ex vivo,* we show that testosterone-related progestins induce the PR target and mediator of PR signaling-induced cell proliferation receptor activator of NF-κB ligand (Rankl), whereas progestins with anti-androgenic properties in reporter assays do not. We develop intraductal xenografts of human breast epithelial cells from 36 women, show they remain hormone-responsive and that progesterone and the androgenic progestins, desogestrel, gestodene, and levonorgestrel, promote proliferation but the anti-androgenic, chlormadinone, and cyproterone acetate, do not. Prolonged exposure to androgenic progestins elicits hyperproliferation with cytologic changes. Androgen receptor inhibition interferes with PR agonist- and levonorgestrel-induced *RANKL* expression and reduces levonorgestrel-driven cell proliferation. Thus, different progestins have distinct biological activities in the breast epithelium to be considered for more informed choices in hormonal contraception.

**Keywords** androgen receptor signaling; breast cancer; hormonal contraception; progestins; xenografts

**Subject Category** Cancer

## Introduction

Breast cancer is the most common cause of cancer-related death among women worldwide, disease incidence continues to increase (https://www.iarc.who.int/wp-content/uploads/2018/07/pr223_E.pdf), and it became the most commonly diagnosed cancer in 2020 (Sung *et al,* 2021). Hence, it is important to better understand its causes and to improve its prevention. Epidemiologic studies show that early menarche, late menopause, and shorter menstrual cycles are associated with increased breast cancer risk (MacMahon *et al,* 1970; Pike *et al,* 1983; Colditz *et al,* 2004). Early menarche and late menopause increase overall lifetime exposure of the breast epithelium to ovarian hormones. The length of the menstrual cycle is determined by the duration of the follicular phase, which varies between women (7–21 days), whereas the luteal phase always lasts 14 days. Hence, the higher breast cancer risk associated with shorter menstrual cycles may relate either to the number of cyclical changes and/or to increased lifetime exposure to specifically luteal phase hormones. During the follicular phase, estrogen levels peak while progesterone levels are very low; the luteal phase is characterized by high progesterone levels in the presence of a smaller peak of estrogen levels. In addition, women are widely exposed to exogenous hormones, both natural and synthetic, in the context of contraception and hormone replacement therapy (HRT), and these exposures have been linked to increased breast cancer risk.

Hormonal contraceptives consist of different progestogens, either on their own as in the minipill and intrauterine devices, or in conjunction with ethinyl estradiol (EE), an estrogen receptor α (ER) agonist, in combined oral contraceptives (OC). They are used by millions of women worldwide and were shown to bestow a 1.24

1  Swiss Institute for Experimental Cancer Research, School of Life Sciences, Ecole Polytechnique Fédérale de Lausanne, Lausanne, Switzerland
2  Centre Hospitalier Universitaire Vaudois, University Hospital of Lausanne, Lausanne, Switzerland
3  International Cancer Prevention Institute, Epalinges, Switzerland
4  The Breast Cancer Now Toby Robins Breast Cancer Research Centre, The Institute of Cancer Research, London, UK
   *Corresponding author. Tel: +41 21 693 07 81; E-mail: cathrin.brisken@epfl.ch
   †These authors contributed equally to this work
   ‡Present address: Wellcome Sanger Institute, Hinxton, UK
   §Present address: Cambridge Institute for Medical Research, University of Cambridge, Cambridge, UK

relative risk (RR) of breast cancer on current users (Collaborative Group on Hormonal Factors in Breast Cancer, 1996). Hormonal contraceptives differ in progestogens, dose, and administration scheme. Progestins can be structurally related to progesterone, testosterone, or spironolactone and interact to different extent with the androgen receptor (AR), the mineralocorticoid receptor (MR), and the glucocorticoid receptor (GR; Schindler et al, 2003). They also vary in bioavailability, efficacy, and have different binding affinities for the sex hormone-binding globulin (SHBG) and corticosteroid-binding globulin (CBG) bound to which these hydrophobic substances travel in the bloodstream (Stanczyk, 2002, 2003).

Hormonal contraception evolved mainly driven by the need to reduce the cardiovascular complications associated with earlier generations of progestins, in particular thrombosis, and new progestins were developed, doses adjusted, and regimens modified (Dhont, 2010). Yet, our understanding of the biological effects of progesterone and different progestins on the breast epithelium is limited. It was proposed that high EE doses of earlier preparations were responsible for the increased RR for breast cancer but a recent study on 1.8 million Danish women showed that current low EE level preparations are also associated with a RR of 1.2 (Mørch et al, 2017), suggesting an important role for the progestin component. Here, we hypothesize that the RR of 1.2 associated with hormonal contraception overall may mask a higher risk associated with some specific progestins, whereas other progestins may be associated with smaller RRs and/or even be protective.

The large spectrum of products, the delay between contraceptive use and breast cancer diagnosis, the difficulty to obtain precise information about contraceptives used from women who frequently changed hormonal contraceptives have made it difficult for epidemiologists to evaluate the impact of individual progestins on breast cancer risk. Yet, there are some indications that indeed some progestins maybe more hazardous than others. Analysis of data from the National Medical Reimbursement Registry of Finland revealed that, specifically, levonorgestrel-releasing intrauterine system (LNG-IUS) despite lower plasma levels was associated with a RR 1.2 for ductal and RR 1.33 for lobular breast cancer (Soini et al, 2016). The nurses' health study with 116,608 female participants aged between 25 and 42 years showed that triphasic EE combined with LNG was associated with a RR of 3.05 (Hunter et al, 2010), suggesting that the use of this particular progestin may increase RR more than others. Based on electronic pharmacy records, OCs involving high-dose estrogen, ethynodiol diacetate, or triphasic dosing with norethindrone were associated with more elevated risks than others (Beaber et al, 2014).

Hence, a better understanding of the biological effects of individual progestins is key to a more informed approach to hormonal contraception, in which high-risk progestins are avoided and progestins that have little effect or are protective are prescribed instead to reduce risk and to offer effective breast cancer prevention to a large population of women.

The biological activities of various progestins have been characterized with cell line-based reporter assays in readily transfectable cell line models. In particular, their differential activities versus the progesterone receptor (PR), androgen receptor (AR), glucocorticoid, and mineralocorticoid receptor have been characterized with reporter assays. A recent side-by-side comparison of various progestins used in OCs revealed cell line- and transfection-related differences (Enfield et al, 2020) pointing to the limited value of such in vitro models for making clinical predictions and the need for better models. Indeed, little is known about the effects of different progestins on the breast epithelium because we lacked adequate models to study hormone action (Özdemir et al, 2018; Brisken & Scabia, 2020). Primary human breast epithelial cells (HBECs) can be readily isolated from reduction mammoplasty specimens but they lose hormone receptor (HR) expression when cultured in vitro. Inhibition of TGFβ signaling was shown to release ER$^+$ HBECs from growth restraint and to enable them to expand in vitro (Fridriksdottir et al, 2015); yet, whether these cells retain PR signaling remains to be evaluated. In sophisticated 3D matrigel cultures, primary HBECs maintain HR expression and proliferate in response to progesterone but expression of functionally important in vivo PR target genes Rankl/Tnfsf11 (Beleut et al, 2010; Mukherjee et al, 2010) and Wnt4 (Brisken et al, 2000; Rajaram et al, 2015) was not induced (Graham et al, 2009).

Here, we test the hypothesis that different progestins have distinct biological effects on the mammary epithelium using a mouse ex vivo model, fresh mammary organoids, which maintain intercellular contacts and remain hormone-responsive (Ayyanan et al, 2011). To assess the effects of prolonged exposure to different progestins on HBECs, we develop "humanized" mouse mammary glands by grafting reporter containing patient-derived breast epithelial cells into the milk ducts and monitoring their growth in vivo. We discern differential effects suggesting that progestins with anti-androgenic activity may be safer to use with regard to breast cancer risk than testosterone-related compounds, in particular, the widely used levonorgestrel.

## Results

### Transcriptional response to PR signaling activation in mouse mammary organoids

We hypothesized that different progestins used for hormonal contraception may differentially affect the breast epithelium and, hence, breast cancer risk. To test this hypothesis, we sought for biological readouts in physiologically relevant models. First, we recurred to using an ex vivo assay for the transcriptional response of the mouse mammary epithelium to hormones. We isolated mammary glands from several inbred adult female mice, treated them mechanically and enzymatically to generate epithelial-enriched organoids, which remain hormone-responsive (Ayyanan et al, 2011), and exposed them either to vehicle or to the stable progesterone mimic promegestone (R5020, 20 nM) for 6 h and performed RNA sequencing (Fig 1A). Three independent experiments were performed, and principal component analysis (PCA) grouped the samples according to treatment (Fig 1B). Under these experimental settings, R5020 stimulation resulted in 162 differentially expressed genes, of which 107 were upregulated and 55 were down modulated (Fig 1C, Appendix Fig S1A). Importantly, 2 genes previously shown to be important mediators of PR signaling in the mammary epithelium, Rankl/Tnfsf11 (19,20) and Wnt4 (Brisken et al, 2000; Rajaram et al, 2015) together with SAM pointed domain containing Ets transcription factor (Spdef) and AR interactor and target (Sood et al, 2007; Cui et al, 2016), were the most significantly induced genes with adjusted P-values < 10E-

20 (Fig 1D). *Rankl* transcripts showed the highest induction by R5020 with 10.07-fold over control samples (Fig 1D), while *Wnt4* was induced 3.1-fold, i.e., to a lesser extent, as observed in previous studies (Tanos *et al*, 2013; Rajaram *et al*, 2015). The *Pgr* transcripts showed the largest down-modulation with 0.2-fold. Together, these observations confirm that PR signaling is intact in this *ex vivo* model and that the functionally important factors, *Rankl* and *Wnt4*, are robust PR target genes in the mouse mammary epithelium.

Gene Set Enrichment Analysis (GSEA) of the differentially expressed genes using C2 and hallmarks MSigDB collections (Subramanian *et al*, 2005) demonstrated that R5020-induced genes are enriched for a stem cell signature, in line with PR's established role in stem cell function (Joshi *et al*, 2010; Axlund & Sartorius, 2012; Rajaram *et al*, 2015) and showed a trend of upregulation of Myc (adj *P*-value 0.054) (Fig 1E). *Estradiol response down* was negatively enriched, and a trend of an increased *androgen response* was observed (adj *P*-value 0.0656) (Fig 1E) suggesting that ER, PR, and AR function may be closely interconnected in the normal breast epithelium as observed in ER$^+$ tumor cells. TGF-β signaling and mitotic spindle were negatively enriched (Appendix Fig S1B), while oxidative phosphorylation, INFγ response, and DNA repair were enriched (Appendix Fig S1C).

**Effect of progestins on *Rankl* and *Wnt4* transcript levels**

Next, we stimulated mouse mammary organoids derived from several *BalbC* females with R5020 or different progestins (Fig 1A). We chose progestins of different generations commonly used in Switzerland, for the first generation: chlormadinone acetate (CMA), for the 2nd generation: levonorgestrel (LNG), for the 3rd generation: 3-ketodesogestrel, the active form of desogestrel (DSG), cyprotherone acetate (CPA), and gestodene (GSN), as well as drospirenone (DSP) for the 4th generation. At 1, 2, or 6 h of stimulation *Rankl* and *Wnt4* transcript levels were determined by RT-PCR. No significant change in *Rankl* and *Wnt4* transcript levels was detected during the first 2 h (Appendix Fig S1D). By 6 h of R5020 stimulation, *Rankl* and *Wnt4* transcript levels increased 22.4 and 5.8-fold over the vehicle-treated control, respectively. DSG, GSN, and LNG-induced *Rankl* transcripts to a similar extent or more with 22.3-, 18.2-, and 27.7-fold, respectively, whereas CMA, CPA, and DSP failed to affect *Rankl* transcript levels at all (Appendix Fig S1D). *Wnt4* transcripts were induced to a lesser extent, by DSG, GSN, and LNG with an 8.5-, 4.9-, and 4.8-fold change, respectively, whereas CMA, CPA, and DSP did not alter *Wnt4* transcript levels (Appendix Fig S1D).

To assess whether the differential regulation of transcript levels depended on the genetic background of the mice, we next stimulated mammary tissue from C57Bl/6 mice. Within 6 h, R5020 induced *Rankl* and *Wnt4* transcripts 10.9- and 4.2-fold, respectively. DSG, GSN, and LNG increased *Rankl* transcript levels 15.5-, 21.5-, and 21.6-fold, respectively. Conversely, CMA, CPA, and DSP failed to increase *Rankl* transcript levels (Fig 1F). *Wnt4* transcripts were induced to a lesser extent, with a 4.2-, 3.6-, and 4.3-fold change induced by DSG, GSN, and LNG, respectively while CMA and DSP did not alter *Wnt4* transcript levels (Fig 1F). CPA induced *Wnt4* 2-fold (p < 0.05) (Fig 1F). A similar result was obtained in an experiment with mammary tissue from NOD. Cg-Prkdc$^{scid}$ Il2rg$^{tm1Wjl}$/SzJ (NSG) mice (Appendix Fig S1E).

The observation that there are two groups of progestins, those that induce *RANKL* and *WNT4* transcripts and those that do not, was confirmed at 24 h of stimulation of BalbC-derived mammary organoids. At this timepoint, *Rankl* and *Wnt4* transcripts were induced 83- and 26-fold by R5020, respectively. DSG, GSN, and LNG increased *Rankl* transcript levels 177.2-, 617.2-, and 253.9-fold and Wnt4 transcript levels 32.2-, 33.5-, and 36.3-fold, respectively, whereas CMA, CPA, and DSP had no effect on either transcript (Fig 1G).

Thus, independent of the genetic background of the tissue donor, GSN, LNG, and DSG like R5020 consistently increase transcript expression of two important PR target genes, *Rankl* and *Wnt4* in mammary gland tissue, whereas CMA, CPA, and DSP do not.

**AR activity is required for *Rankl* induction by R5020 and LNG**

We noticed that the inducers of *Rankl* and *Wnt4* transcripts, DSG, GSN, and LNG are 13-ethylgonanes, which are structurally related to testosterone and have all been reported to show androgenic activity in various assays, whereas CPA, CMA, and DSP had shown anti-androgenic activity (Fuhrmann *et al*, 1995; Muhn *et al*, 1995; Bouchard, 2005; Mishell, 2008; Schneider *et al*, 2009; Africander *et al*, 2011; Louw-du Toit *et al*, 2017). To test whether the anti-androgenic activity of CPA, CMA, and DSP may be responsible for the inhibition of *Rankl* and *Wnt4* transcript induction, we treated mammary organoids *ex vivo* with R5020 and the nonsteroidal AR antagonists enzalutamide and bicalutamide. Enzalutamide (100 μM) abrogated *Rankl* induction by R5020 (*P* < 0.05) but had no effect on *Wnt4* induction (Fig 1H). Furthermore, induction of *Rankl* by the widely used androgenic progestin, LNG, was significantly reduced by enzalutamide (100 μM) (*P* < 0.05), whereas *Wnt4* induction again was not affected (Fig 1I). At 10 μM, a trend was observed but this failed to reach significance with *n* = 3, likely due to the heterogeneous biological assay and lack of a pretreatment period. Similarly, bicalutamide (100 μM) abrogated *Rankl* induction by R5020 (*P* < 0.05) but had no significant effect on *Wnt4* induction (Appendix Fig S1F). Thus, AR activity is required for R5020 and LNG-induced *Rankl* mRNA expression.

**Transcriptional response of the human breast epithelium to PR signaling**

Having ascertained with mammary tissue from inbred mice that androgenic and anti-androgenic progestins have different effects on the expression of important PR target genes, we set out to assess the transcriptional response to PR signaling in the human breast epithelium. In view of our goal to perform *in vivo* experiments with the natural ligand progesterone, we compared the effects of progesterone to those of its stable analog R5020 which is commonly used for *in vitro* experiments on breast epithelial cells. For this, we used an *ex vivo* approach with human breast tissue similar to the mouse organoid technique used above that maintains hormone responsiveness (Tanos *et al*, 2013; Sflomos *et al*, 2015). In brief, 3 fresh mammoplasty tissue from women with follicular phase progesterone levels were cut into 1 mm$^3$ pieces and aliquoted for digestion with collagenase and concomitant exposure to either vehicle, R5020, or progesterone for 14–18 h. Tissue microstructures that contain epithelial cells, immune cells, and fibroblasts in varying

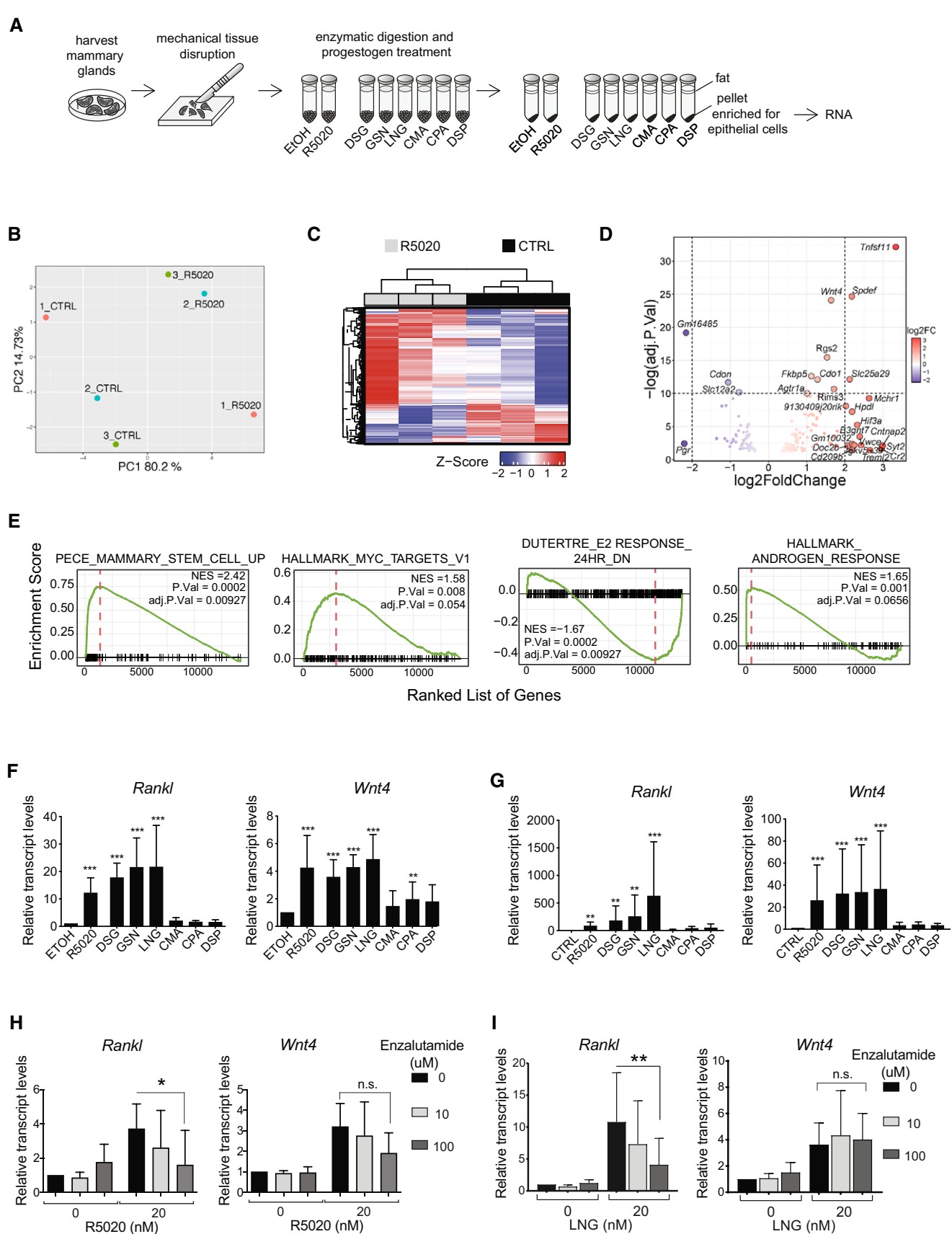

**Figure 1.**

**Figure 1. *Ex vivo* stimulation of murine mammary organoids with different PR agonists.**

A Experimental scheme showing the preparation of organoids from mouse mammary glands and *ex vivo* stimulation with different compounds.

B PCA plot of global gene expression profiles from ethanol (CTRL) and R5020-treated murine mammary organoids.

C Unsupervised hierarchical clustering of the 500 most variable genes across samples. Normalized expression levels were scaled to Z-scores for each gene.

D Volcano plot showing differentially expressed genes (adj. *P*.Val < 0.05) between ethanol- and R5020-treated mouse mammary organoids, $n = 3$. Genes with $\log_2(\text{FC}) > 2$ are highlighted in red, genes with $\log_2\text{FC} < -2$ in blue. Names of genes with $-\log_{10}$ (adj.*P*.Val) > 10 and abs($\log_2$(FC)) > 2 are indicated.

E GSEA showing enrichment of pathways differentially regulated in mouse mammary organoids upon R5020 stimulation, total number of genes = 14,244. The red dashed line indicates NES: Normalized Enrichment Score.

F Bar graphs showing *Rankl* and *Wnt4* transcript levels, relative to 36B4 mRNA expression in mammary organoids derived from C57Bl/6 females 6 h after stimulation with 20 nM R5020 ($n = 12$), DSG ($n = 6$), GSN ($n = 6$), LNG ($n = 6$), CMA ($n = 6$), CPA ($n = 6$), or DSP ($n = 6$).

G Bar graphs showing relative *Rankl* and *Wnt4* transcript levels normalized to *36B4* mRNA expression in mammary organoids derived from Balb6/C females 24 hours after stimulation with different progestins, $n = 3$ independent experiments.

H, I Bar graphs showing relative *Rankl* and *Wnt4* transcript levels normalized to *36B4* expression in C57Bl6-derived mammary organoids treated for 6 hours with R5020 (H) or LNG (I) and either 10 μM or 100 μM enzalutamide, $n = 3$ independent experiments.

Data information (F–I): Data are shown as means ± SD. *P*-values were estimated using a random-intercept linear model applied to ΔCt values followed by Dunnett test. *$P < 0.05$, **$P < 0.01$, ***$P < 0.001$, n.s: not significant.

proportions were recovered. To reduce interpatient variation as far as it relates to different ratios of various cell types, at the end of the treatment, the tissue microstructures were dissociated to single cells and enriched for epithelial cells by FACS sorting with antibodies against the epithelial cell surface marker EpCAM (Fig 2A). The EpCAM$^+$ cells were processed for RNA sequencing. PCA showed clustering according to treatment, with progesterone and R5020-exposed samples grouping together (Fig 2B). Progesterone affected expression levels of 65 genes, whereas R5020 affected 112 genes, including *RANKL* (adj.*P* < 0.05) (Appendix Fig S2A). More than 90% of the progesterone-induced gene expression changes were induced by R5020 (Fig 2C). Log-scaled normalized values (VSD, i.e., variant stabilized gene expression values) for *RANKL* and *WNT4* revealed increased expression of these 2 PR target genes (*P* < 0.01) by both progesterone and R5020 (Fig 2D). Correlation analysis demonstrated that genes affected by both P4 and R5020 correlated in extent of induction (Fig 2E). R5020 induced higher-fold changes as indicated by the estimated slope of the curve being > 1 (*P* < 0.0001). GSEA revealed an enrichment for MYC targets and effects on estrogen and androgen responses as observed in the mouse epithelium in both progesterone and R5020 treated human breast tissue microstructures (Fig 2F). Thus, natural progesterone and R5020 share target genes, with R5020 showing more robust effects on gene expression of EpCAM+ breast cells, as shown previously for T47D cells (Bray *et al*, 2005).

To assess the similarity in hormone response between mouse and human breast cells, we compared the genes differentially expressed in response to R5020 in mouse organoids and human EpCAM$^+$ cells. When considering the genes whose adjusted *P*-value was < 0.05 in both conditions, *RANKL* and *WNT4* were uniquely upregulated and *WNT5A* and *FOXA1* were the two only genes that were down-regulated. (Fig 2G). *In situ* hybridization analysis of human breast tissue sections using RNAscope showed that *RANKL* and *WNT4* transcripts were largely co-expressed in a subset of luminal cells (Appendix Fig S2B). GSEA of the R5020-induced genes in mouse and human *ex vivo* samples revealed that RNA and mRNA metabolism, as well as translation reactome, are enriched in addition to MYC targets (Fig 2H). Notably, androgen response-related genes were also significantly enriched in both species, suggesting that hormone receptor crosstalk is an evolutionarily conserved phenomenon (Fig 2H). In both species, HDAC3 targets and genes down-modulated by E2 were negatively enriched (Fig 2H). Thus,

the PR signaling gene signatures largely overlap between species in spite of different individual target genes.

## Intraductal xenografts of primary HBECs

Next, we sought to assess the response of HBECs to prolonged exposure to different progestins *in vivo*. To this aim, we "humanized" the mouse mammary glands. Freshly isolated HBECs were infected with GFP-luciferase2-expressing lentiviruses and injected into the milk ducts of adult NSG females (Sflomos *et al*, 2016) (Fig 3A). *In vivo* bioluminescence monitoring of the engrafted mice showed that human cells were able to establish themselves in mouse mammary ducts and to grow. On average, radiance increased 12-fold over 100 days (Fig 3B). At sacrifice, fluorescence stereomicroscopy revealed GFP signal in multiple ducts of the injected mammary glands (Fig 3C). Whole mount stereomicroscopy of xenografted glands revealed the mouse milk duct system and showed that some ducts are dilated (Fig 3D, arrow). H&E stained histological sections revealed ducts lined by HBECs, which have larger lumina than their murine counterparts (Fig 3E). Immunofluorescence (IF) with a human-specific antibody for the luminal marker CK19 unequivocally confirmed the presence of HBECs (Fig 3F). Similarly, immunohistochemistry (IHC) with an anti-human CK19 antibody revealed that the HBECs are largely continuous with the mouse mammary epithelium and appear to replace it (Fig 3G and H).

To assess whether the engrafted HBECs retain HR expression, we co-stained adjacent sections of xenografted glands with antibodies specific for human E-Cadherin (hECAD) and the different steroid receptors. A subset of HBECs were positive for ER, PR, and AR showing the same cluster pattern of receptor expression as the human breast epithelium (Fig 3I). To assess whether the human cells respond to hormones, we impregnated recipients with HBECs in their mammary glands. Human epithelial cells were readily detected in histological sections by anti-human CK7 staining, these revealed that the lumen was more distended suggesting secretory activity (Fig 3J). Semi-quantitative RT-qPCR analyses using human-specific primers showed that the human milk genes α-lactalbumin, κ-casein, and α-S1-casein were readily detected in pregnant females while the transcripts were below detection limit (CT ≥ 40) in virgin hosts. (Fig 3K). Thus, normal HBECs grow in mouse mammary ducts, preserve HR expression, and remain hormone-responsive.

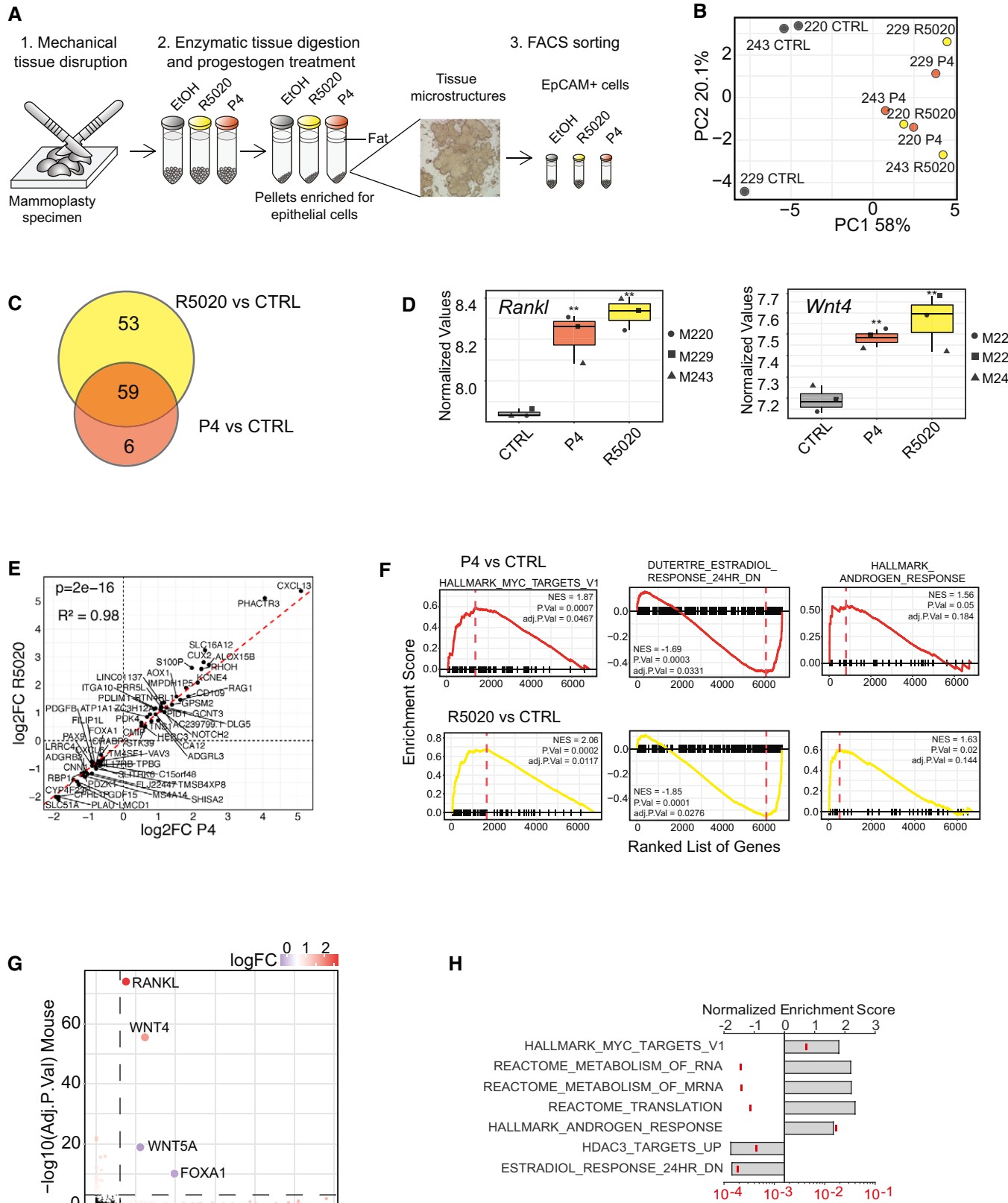

**Figure 2.**

**Figure 2.  *Ex vivo* stimulation of human breast tissue microstructures with different PR agonists.**

A   Experimental scheme showing the preparation of human breast tissue microstructures from reduction mammoplasty specimens, the *ex vivo* stimulation with R5020 and progesterone and epithelial purification by FACS sorting based on EpCAM expression.
B   PCA plot showing global gene expression profiles of EpCAM$^+$ cells isolated from ethanol-, P4-, and R5020-treated human breast tissue microstructures, $n = 3$.
C   Venn diagram showing differentially expressed genes (adjusted *P*-value < 0.05) in response to either P4 or R5020 relative to CTRL samples.
D   Boxplot showing log-scaled variant stabilized gene expression values for *RANKL* and *WNT4* in human tissue microstructures, $n = 3$. Shown are the non-corrected *P*-values estimated by DESEQ2. **$P < 0.01$. Vertical lines outside the box end at maximum and minimum values, upper and lower borders of the box represent lower and upper quartiles, and line inside the box identifies the median.
E   Scatter plot showing a linear relationship between the $\log_2 FC$ of shared differentially expressed genes between R5020 and P4 relative to CTRL samples shown in (B). Coefficients, *P*-value on the estimated coefficient, and adjusted R squares were calculated by linear regression model (lm function in R). The dashed red line shows the diagonal.
F   GSEA of differentially expressed genes between P4 (red) and R5020 (yellow) relative to CTRL samples. The red dashed lines indicate NESs.
G   Scatter plot showing the genes which are commonly altered upon R5020 stimulation (adj. *P*-value < 0.05) in both mouse and human organoids relative to their CTRL samples. Color scale as a function of the averaged human and mouse logFC values. The red dashed lines represent an adj. *P*-value threshold of 0.001.
H   Bar graph showing GSEA of genes altered in both mouse and human EpCAM+ cells upon R5020 stimulation of tissue microstructures. Bars define the averaged NES between *ex vivo* R5020-treated human and mouse tissue, ticks indicate averaged *P*-values given by GSEA.

## Progestin pellets and plasma levels

To compare the effects of progesterone and different progestins on HBECs *in vivo*, we infected single cell suspensions obtained from fresh reduction mammoplasty specimens with GFP-luciferase2-expressing lentiviruses. The infected cell mixture was grafted intra-ductally to up to 5 glands of as many recipients as possible. The engraftment and growth of the xenografted cells were monitored weekly by *in vivo* imaging. When radiance reached > 10E07 on average per mammary gland, 60-day-release pellets were implanted subcutaneously, usually within 3–10 weeks after cell injection. At sacrifice, blood was collected and plasma levels of the administered compounds as well as endogenous steroid hormones were determined by liquid chromatography-mass spectrometry (LC-MS) (Laszlo *et al*, 2019).

In mice engrafted with 20 mg progesterone pellets, plasma progesterone ranged from 0.9 to 16.9 ng/ml with an average of 9.1 ng/ml. These levels are comparable to those observed in humans during luteal phase (Kratz *et al*, 2004) and are 4.9 times higher than the average progesterone levels in control mice, which showed estrous cycle-related variation (Fig 4A).

Next, plasma levels of different progestins in mice implanted with subcutaneous pellets were assessed. GSN treated mice showed levels similar to those detected in the blood of women in our study and within the range of reported $C_{max}$ for different doses of GSN containing pills (Stanczyk, 2002) (Fig 4C). LNG plasma levels in experimental animals were similar to those detected in plasma of women in our cohort who were using LNG containing OCs, while women using LNG-IUD had lower LNG plasma levels (Fig 4D); the reported $C_{max}$ for 150 µg and 75 µg containing pills (Stanczyk, 2002) was in similar range (Fig 4D). Plasma levels of DSG, CMA, CPA, and DSP in treated mice were also similar to those detected in the plasma of women in our cohort but tended to be lower than reported $C_{max}$ (Stanczyk, 2002) (Fig 4B and E–G). Thus, plasma levels of progestins in mice treated with slow release pellets are comparable to those found in samples from our patient cohort assessed by the same LC-MS protocol.

Next, we assessed whether progesterone and progestins differentially affected endogenous hormone levels in mice. Plasma 17-β-estradiol levels were not affected by any of the treatments (Fig 4H). In control mice, progesterone levels showed estrous cycle-related variation. Cycles were suppressed by DSG ($P < 0.01$), GSN

($P < 0.05$), and DSP ($P < 0.001$); LNG and CPA did not affect the endogenous progesterone levels and CMA-treated mice showed a tendency toward higher average level of progesterone compared to untreated mice (Fig 4H). Plasma testosterone levels were not altered by any treatment with the exception of DSP, which decreased them (Fig 4H). Thus, prolonged exposures to progestins differentially affect endogenous progesterone plasma levels and DSP decreases testosterone plasma levels in host mice.

### *In vivo* effects of progestins on HBEC proliferation

Next, the *in vivo* growth of HBEC xenografts stimulated with progesterone or different progestins was analyzed. A total of 36 different mammoplasty samples (Appendix Table S1) were engrafted to several glands of multiple mice, and up to 7 different conditions were tested on any one patient's xenografts (Appendix Table S2). In establishing this approach, xenograft growth in individual glands was measured by bioluminescence over 60 days of treatment to ensure enough data points were collected to reach statistically significant results with this highly variable *in vivo* readout. Using log-transformed raw radiance values, we first determined whether the course of the control and treated glands differed significantly. In addition, the fold change radiance of treated versus untreated samples at endpoint was calculated (Fig 5A–C). Progesterone increased the growth rate ($P < 0.001$); at endpoint, relative radiance was increased 1.3-fold over control ($P < 0.01$) (Fig 5A). All 3 testosterone-related progestins, DSG, GSN, and LNG, increased growth ($P < 0.001$) and relative radiance at endpoint 2-, 2.2-, and 2.1-fold, respectively ($P < 0.001$) (Fig 5B). Neither CMA nor CPA had any effect on the growth of the engrafted HBECs. DSP treatment failed to affect the course of the *in vivo* growth, but at endpoint relative radiance was increased 1.4-fold ($P < 0.05$) (Fig 5C).

Using human-specific primers, we determined mRNA levels of prostate-specific antigen (PSA) or Kallikrein 3 (*KLK3*), considered a marker of androgenic activity in prostate and breast cancer cells (Magklara *et al*, 2002; Attardi *et al*, 2004). Consistent with AR and PR sharing a common hormone response element (Claessens *et al*, 1996), *KLK3* was induced upon progestin stimulation. Expression was induced more strongly and was more significantly increased by the androgenic than by the anti-androgenic progestins (Appendix Fig S3). This is in agreement with previous findings that

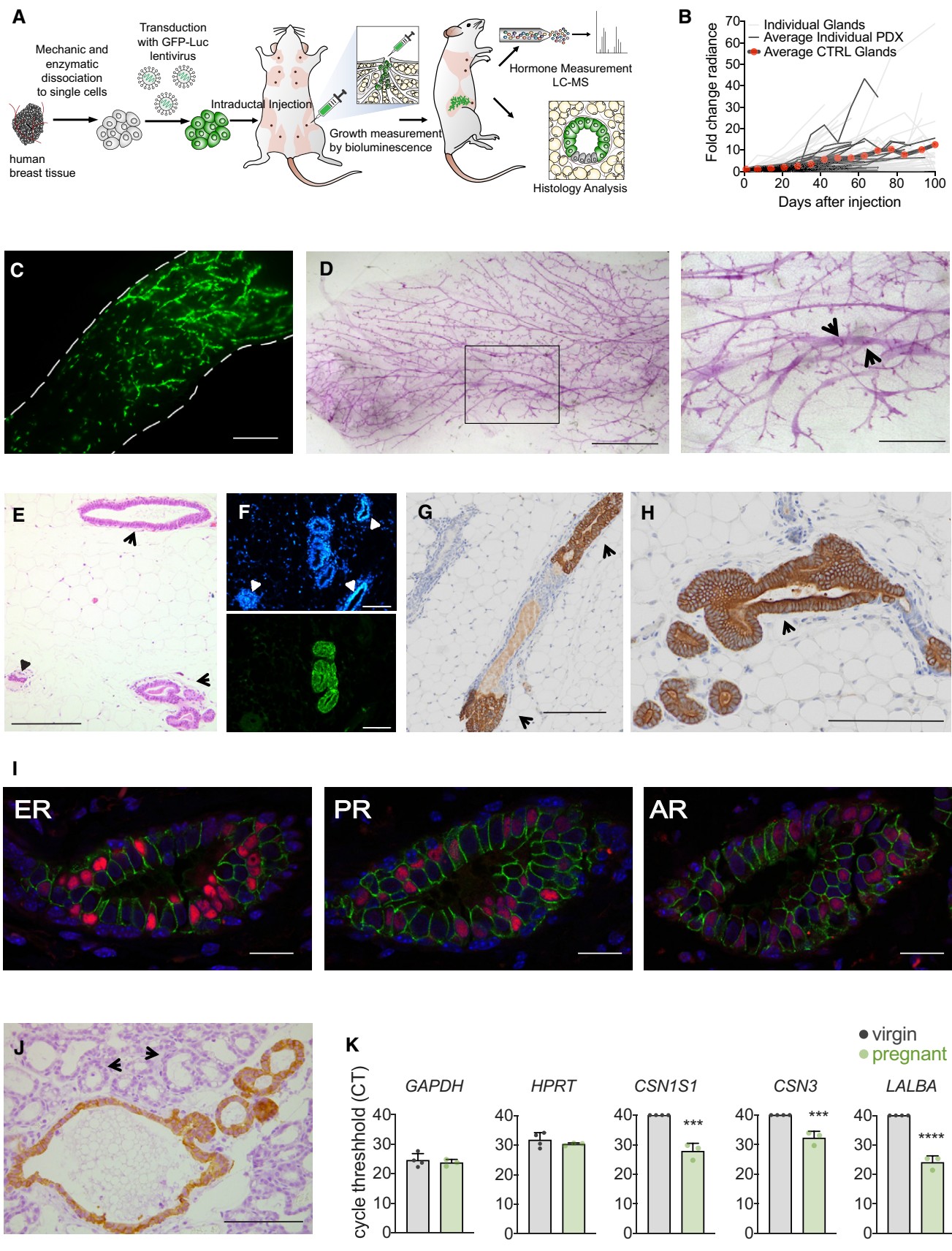

**Figure 3.**

**Figure 3.  Human breast epithelial cell xenografts are hormone-responsive.**

A       Scheme of the mouse intraductal xenograft (MIND) approach. Freshly isolated human breast tissue is dissociated to single cells, which are lentivirally transduced and injected into the milk ducts via the teat. *In vivo* growth is monitored by weekly luminescence measurements. At endpoint, xenografted mammary glands and blood are collected for further analyses.

B       Spaghetti plot showing fold change radiance over time, individual xenografted mammary glands (light gray), averaged control samples from individual xenografts (dark gray), average values of control samples from all xenografts (black, red dots), $n = 12–224$ xenografted glands per time point.

C       Fluorescence stereo micrograph of inguinal mammary gland 10 weeks after injection of GFP labeled HBECs. Dashed lines highlight the perimeter of the mammary fat pad. Scale bar, 3 mm.

D       Stereo micrographs of whole mounted mammary glands 12 weeks after injection of HBECs. Scale bar, 2 mm. Inset, arrows point to mouse ducts distended by the presence of HBECs. Scale bar, 0.8 mm.

E       H&E stained section of xenografted gland 83 days after injection showing HBECs that distend a mouse duct (arrow), small mouse ducts are shown (small arrow heads). Scale bar, 0.1 mm.

F       Fluorescence micrograph of histological section of an intraductally xenografted mammary gland stained with DAPI (top) and immunofluorescence with anti-hCK7 (bottom). Arrows point to mouse ducts devoid of human cells. Scale bar, 0.1 mm.

G, H   IHC for hCK19, in brown, of cross-sectioned "humanized" mouse milk duct mammary glands 106 days after injection. Arrows point to mouse ducts colonized by human cells. Scale bars, 100 µm (G) and 150 µm (H), respectively.

I       Representative co-IF micrographs of a cross-sectioned "humanized" mouse milk duct 5 months after intraductal injection for AR, ERα, and PR, in red, and human E-CAD, in green. Scale bar, 50 µm.

J       IHC for hCK7, in brown, of cross-sectioned "humanized" mouse milk duct mammary glands at day 20 of pregnancy. Arrow heads point to mouse alveoli, stars highlight lumina filled with secretions. Scale bar, 100 µm.

K       Bar graphs showing the threshold cycles of semi-quantitative RT-PCR with human-specific primers on RNA from virgin ($n = 4$) and pregnant ($n = 3$) mice xenografted with human cells.

Source data are available online for this figure.

*KLK3* expression is under the control of both androgens and progestins in T47D cells (Zarghami *et al*, 1997).

Next, we sought for a readout of *in vivo* cell proliferation different from radiance. Analysis of the xenografted glands for Ki67 expression by IHC failed to reveal correlation with the *in vivo* growth measurements after the 60-day treatment. As the slope of the growth curves tends to be higher in the first 3 weeks than later during the treatment, we analyzed mice engrafted with HBECs after 3 weeks of treatment with LNG. Radiance measurements confirmed that LNG stimulated cell proliferation (Fig 5D), and at endpoint, the expression of *MKI67* at both protein level as assessed by IHC (Fig 5E and F) and mRNA level (Fig 5G) was increased. Moreover, LNG-treated samples also showed increased *KLK3* transcript expression suggesting that AR signaling is active (Fig 5H). Given that MKI67 expression could replace the luciferase measurements as a readout, we used a modified experimental setup for a second round of experiments. We derived HBECs from viably frozen mammoplasty samples (Appendix Table S3) and engrafted them to different mammary glands of NSG females without viral infection. Four months later, the 24 recipient females, each engrafted with 5 different mammoplasties received pellets containing either vehicle, progesterone, or 1 of 6 progestins for 21 days. At sacrifice, xenografted glands were histologically sectioned and cell proliferation was quantified using co-IF with anti-MKI67 and anti-hECAD antibodies. For one of the mammoplasties, we were able to obtain measurements for all experimental conditions (Fig 5I). This showed that progesterone (P < 0.001), DSG (P < 0.01), GSN (P < 0.001), and LNG (P < 0.01), significantly increased cell proliferation while CPA (P < 0.05) decreased it, whereas CMA and DSP had no effect (Fig 5J). For the remaining mammoplasties, between 3 and 8 different experimental conditions were successfully quantified (Appendix Table S4). In line with the previous findings, the overall analysis showed that progesterone (P < 0.001), DSG (P < 0.0001), GSN (P < 0.0001), and LNG (P < 0.0001) increased cell proliferation. CPA (P < 0.0001) decreased proliferation,

whereas CMA and DSP had no effect (Fig 5K). As AR signaling increases AR protein levels, we performed co-IF for AR and hECAD as another means to assess AR signaling activity. Progesterone (P < 0.001), CPA (P < 0.0001), and DSP (P < 0.01) decreased the percentage of AR+ cells, whereas DSG (P = 0.12), GSN (P < 0.0001), and LNG (P < 0.0001) increased the AR index and CMA had no effect (Fig 5L). Semi-quantitative RT-PCR for *KLK3* showed transcripts levels were increased by the androgenic progestins, DSG (P < 0.01), and GSN (P < 0.001) (Fig 5 M). Thus, the testosterone-related progestins, DSG, GSN, and LNG stimulate cell proliferation in the human breast epithelium *in vivo* and induce expression of *KLK3*, whereas the tested anti-androgenic progestins do not. Conversely, CPA even showed an antiproliferative effect in the 21-day treatment experiments.

**Long-term stimulation of the human breast epithelium with different progestins**

To test whether the increased cell proliferation observed with androgenic progestins may elicit morphologic changes in the breast epithelium, we subjected 10 mice engrafted with 5 different mammoplasties to treatment with either vehicle, DSG, GSN, or LNG for 8 months. All the engrafted glands were processed for histological analysis and H&E stained. In control females, the 5 mammoplasty-derived HBECs showed only limited growth of human cells (Fig 6A, Appendix Fig S4), in 2 of them occasional foci of cell proliferation (Fig 6B) were detected. In xenografts stimulated with either DSG, GSN, or LNG, 3 or 4 grafts showed diffuse and in some cases marked acinar proliferation (Fig 6C, Appendix Fig S4). In addition, variable ductal dilatations up to cyst formation (Fig 6D) with apocrine epithelial changes and focally complex apocrine hyperplasia (Fig 6E and F) were observed. Thus, in a small set of samples examined, the prolonged treatment with androgenic progestins increased the incidence of morphologic proliferative changes that are part of early precursor stages to malignancy in the human breast epithelium.

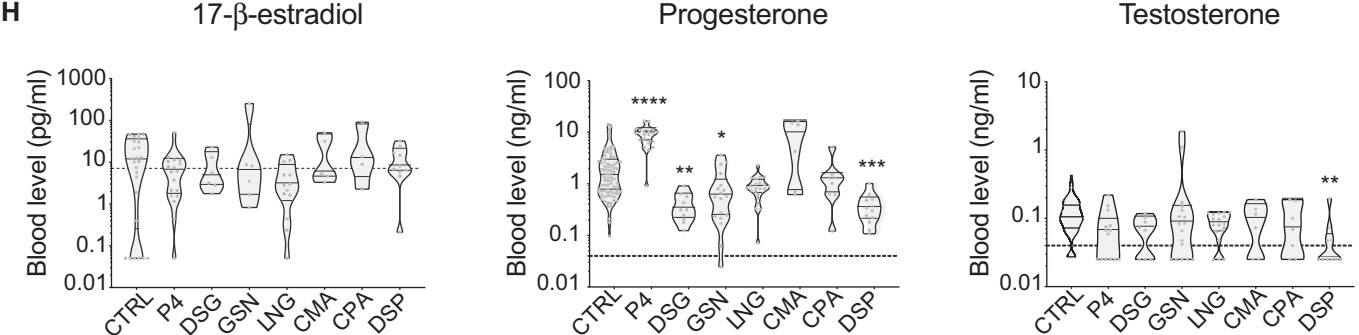

**Figure 4. Validation of progestogen pellets.**

A    Progesterone plasma levels in control ($n = 79$) and progesterone-treated ($n = 20$) mice at sacrifice as measured by LC-MS, shown as means ± SD.

B–G  Progestin plasma levels in treated mice ($n = 10, 14, 18, 7, 6, 10$) and in women ($n = 20, 11, 3, 4, 6, 8$) on hormonal contraceptives measured by LC-MS, shown as means ± SD, as well as values reported in the literature ($C_{max}$). The active form of DSG, 3-ketodesogestrel was measured.

H    Violin plots showing plasma levels of 17-β-estradiol, progesterone, and testosterone in individual xenografted NSG females measured by LC-MS; $n = 7$–106 per treatment. Dotted line indicates lower limit of detection. Statistical significance was tested by non-parametric Kruskal–Wallis test, followed by Dunn's test. *$P < 0.05$, **$P < 0.01$, ***$P < 0.001$, ****$P < 0.0001$.

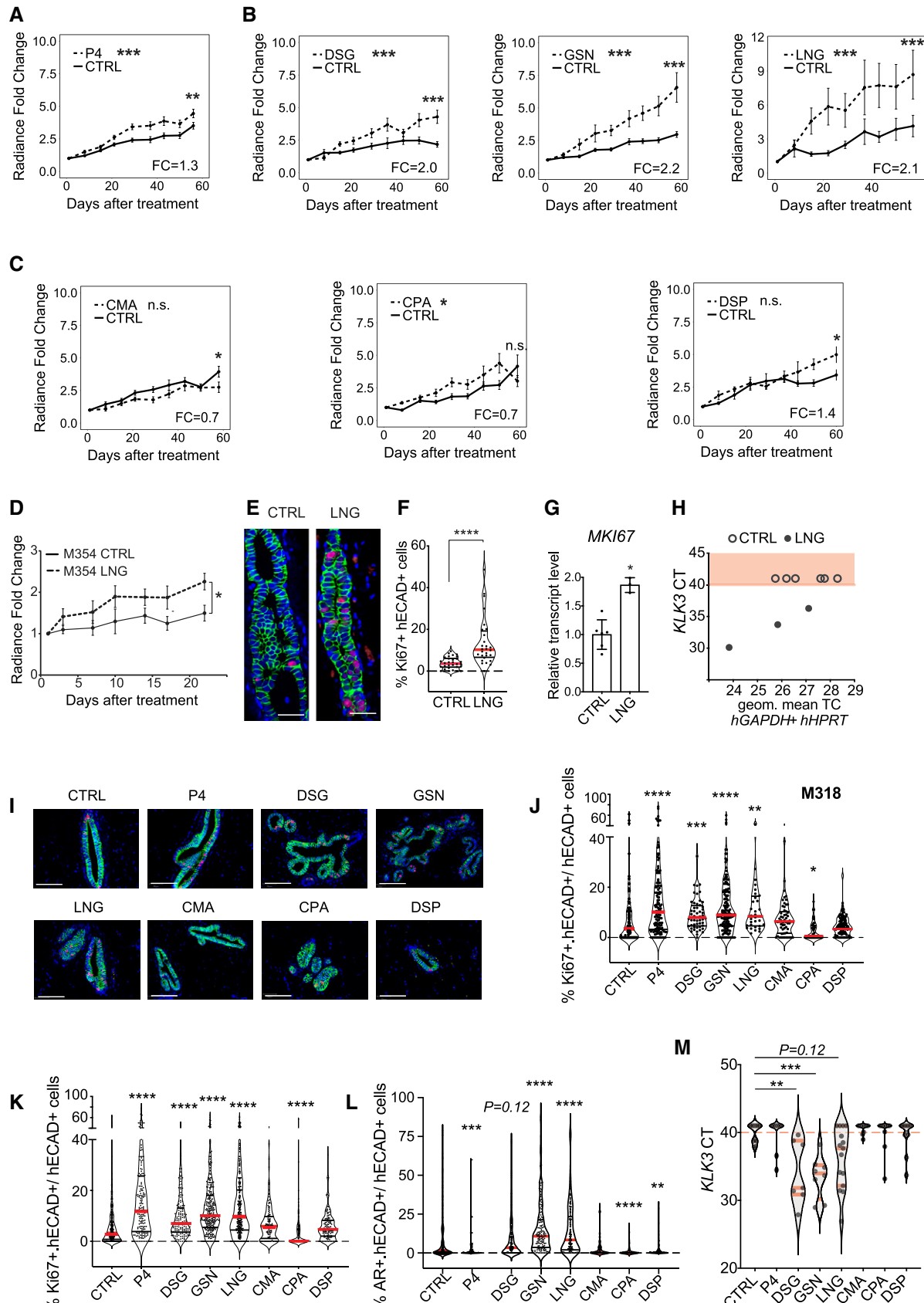

**Figure 5.**

◀

**Figure 5.   Effects of PR agonists on human and mouse epithelial cells growth.**

A–C   Line charts showing *in vivo* growth of HBECs as measured by radiance after implantation of either control or different progestogen-containing pellets. Points show means of radiance in individual glands ± SEM. Statistical significance was tested by fitting a mixed effect linear model with random effects to the log10-transformed data; $n = 87$ and 64, respectively, for control and progesterone, $n = 34$, respectively, for both control and DSG, $n = 59$ and 39, respectively, for control and GSN, $n = 70$ and 79, respectively, for control and LNG, $n = 33$ and 20, respectively, for control and CMA, $n = 28$ and 30, respectively, for both control and CPA, $n = 48$ and 35, respectively, for control and DSP and shown by asterisks next to legend. To test differences at endpoint, Wilcoxon rank-sum test was applied on log-transformed fold change values, asterisks at endpoints of graphs.

D   Graph showing *in vivo* growth of HBECs from mammoplasty M354 as measured by radiance upon either control ($n = 9$) or LNG ($n = 8$) treatment. Points show means of radiance in individual glands ± SEM. Statistical significance of difference at endpoint was tested by Student's *t*-test.

E   Representative micrographs showing co-IF with anti-KI67 (red), and anti-hECAD (green) of xenografted milk ducts after 21 days treatment with vehicle or LNG. Scale bar, 50 μm.

F   Violin plot showing the percentage of KI67 and hECAD double+ cells of total hECAD+ cells, dots represent individual sectors counted in 3 different glands, control ($n = 30$) or LNG ($n = 28$), median (red), statistical significance was assessed by non-parametric Mann–Whitney test.

G   Bar plot showing relative transcript levels of *MKI67* in xenografted glands from control ($n = 6$) and LNG-treated ($n = 3$) mice, Student's *t*-test.

H   Graphs showing threshold cycle number of *KLK3* plotted over housekeeping genes, semi-quantitative RT-PCR performed with human-specific primers on RNA extracted from control ($n = 6$) and LNG-treated ($n = 3$) xenografted glands.

I   Representative micrographs showing co-IF with anti-KI67 (red) and anti-hECAD (green) of histological sections from glands xenografted with HBECs from mammoplasty M310 exposed to vehicle or progestins. Scale bar, 50 μm.

J   Violin plot showing the percentage of KI67 and hECAD double+ cells of total hECAD+ cells in M310 derived HBECs, dots represent individual sectors counted in different glands, median (red), CTRL ($n = 180$), P4 ($n = 134$), DSG ($n = 52$), GSN ($n = 161$), LNG ($n = 32$), CMA ($n = 56$), CPA ($n = 42$), DSP ($n = 100$). Statistical significance was assessed by fitting a generalized linear mixed model with gamma distributions using the CTRL group as reference.

K   Violin plot showing the percentage of KI67 and hECAD double+ cells of total hECAD+ HBECs derived from 11 different patients upon 21 days of CTRL ($n = 360$), P4 ($n = 180$), DSG ($n = 147$), GSN ($n = 298$), LNG ($n = 284$), CMA ($n = 115$), CPA ($n = 156$), DSP ($n = 123$) treatment. Red lines show medians.

L   Violin plot showing percentage of AR+ intraductally-engrafted HBECs derived from 11 different patients upon 21 days of CTRL ($n = 258$), P4 ($n = 97$), DSG ($n = 173$), GSN ($n = 159$), LNG ($n = 168$), CMA ($n = 68$), CPA ($n = 76$), DSP ($n = 135$) pellets. Statistical significance was assessed by fitting a generalized linear mixed model with gamma distributions, with batches and patients as random variables and CTRL as reference, median in red. J–L dashed line shows 0%.

M   Violin plot showing cycle threshold (CT) of *KLK3* transcripts upon progestin treatment, as assessed by qRT-PCR, in CTRL ($n = 13$), P4 ($n = 10$), DSG ($n = 7$), GSN ($n = 8$), LNG ($n = 15$), CMA ($n = 12$), CPA ($n = 12$), DSP ($n = 12$) conditions. Statistical analysis performed by non-parametric Kruskal–Wallis test followed by Dunn's test with CTRL as reference. Red dashed line corresponds to CT40 considered detection limit.

Data information: *$P < 0.05$, **$P < 0.01$, ***$P < 0.001$, ****$P < 0.0001$ n.s. not significant.

## AR signaling is required for LNG-induced proliferation of HBECs

To test whether AR activity is required for HBEC proliferation induced by androgenic progestins, specifically the widely used LNG, we treated mice-bearing xenografted mammary glands with LNG either in presence or absence of the AR antagonist enzalutamide. LNG exposure increased the proliferation rates in 2 of 3 PDXs. In both of these, enzalutamide decreased the proliferative effects of LNG (Fig 7A–C) suggesting that AR is required for LNG-induced proliferation. The third mammoplasty, which did not show a proliferative response to LNG, came from a patient in whose blood we detected the progestin nomegestrol acetate (NOMAC) (Fig 7A–C).

To further test the requirement for AR in the context of progestin-driven proliferation, we took a genetic approach and down modulated AR expression in freshly isolated HBECs using a *GFP-luciferase2-shAR* or *GFP-luciferase2-sh* scramble expressing lentivirus. *In vivo* bioluminescence monitoring of the engrafted mice showed that down regulation of AR expression reduced LNG-induced radiance by 22% within 60 days (Fig 7D). Double IF with anti-GFP and anti-AR antibodies and analysis by image quantification confirmed the down regulation of AR protein levels in the *shAR* infected cells identified based on GFP co-expression (Fig 7E and F). Taken together, these results suggest that AR may be at least partially required for HBEC proliferation induced by the androgenic progestin LNG.

## Discussion

Different hormonal formulations are taken by millions of women worldwide, both in the context of contraception and hormone replacement therapy. Exploiting *ex vivo* and new intraductal xenograft models to study hormone action, we provide evidence that different progestins elicit different biological effects on the breast epithelium. This has important clinical and public health implications as our study suggests that hormone-based drugs may differentially affect breast cancer risk and that a more informed approach to hormonal contraception may decrease the incidence of breast cancer. By avoiding hormonal contraceptives containing progestins, which stimulate cell proliferation of the breast epithelium such as the androgenic LNG, GSN, and DSG, women may reduce their RR for breast cancer. It is conceivable that other hormonal contraceptives, in particular those containing CPA, or DSP, may have preventative effects.

There are various limitations to our study. Firstly, the hormonal milieu of HBECs in mice is not identical to that in humans; 17-β-estradiol and testosterone levels are lower and progesterone levels are higher. This may attenuate the effects of progesterone in our model because 17-β-estradiol induces PR expression (Haslam, 1988) and sensitizes the tissue to progesterone action and higher progesterone levels further reduce PR expression. This mimics some progestins which are administered as depots but lacks the break in progestins exposure in oral contraception, when women are exposed to placebo during standard 28 days oral OC cycle. However, the results of the 21-day experiments, which ostensibly mimic a cycle of hormonal contraception, are concordant with those of the 60-day-treatments. Future studies can be adapted to mimic more closely the different clinical scenarios. Thirdly, when women take OCs, blood progestins level undergo fluctuations that follow the dynamics of daily intake. The hormone pellets we implanted in mice lacked this progestin cyclicity but provided constant

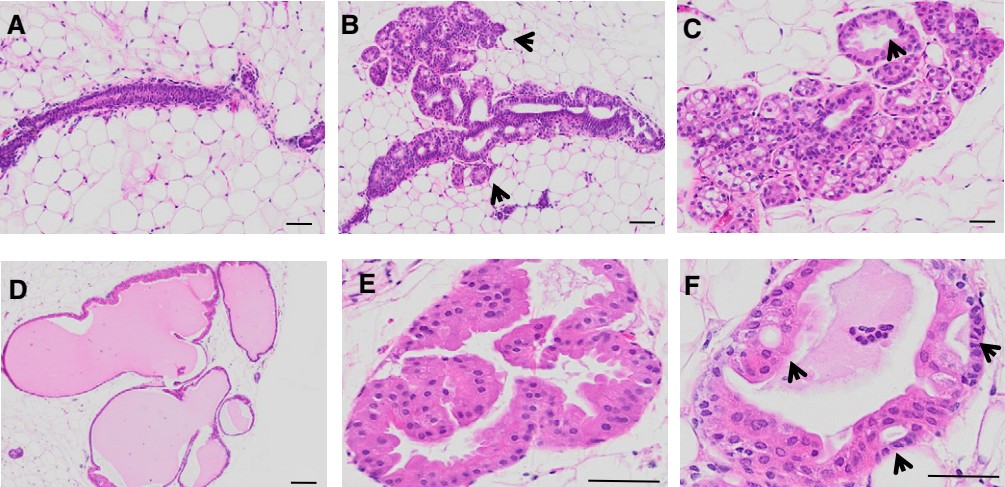

**Figure 6. Effects of long-term stimulation with androgenic progestins on human breast epithelium.**

A–F   Micrographs of H&E stained tissue sections from mouse mammary glands engrafted with HBECS and exposed to LNG or vehicle for 6 months. In control mice (A, B) most ducts have a narrow lumen and a thin wall due to limited growth of human cells (A); a few mildly dilated ducts exhibit a cellular proliferation made out of multiple small acini burgeoning at the external side of their wall (B, arrows). In LNG-treated mice (C–F) the peripheral acinar proliferation is diffuse, marked in some foci (C), and associated with a variable degree of duct dilatation from mild (C, arrow) up to true cysts formation (D). Proliferating human epithelial cells showing apocrine changes (e.g., abundant eosinophilic cytoplasm and central round nuclei) partly fill the ductal lumen with a complex architecture of micropapillae (E) and bridges (F, arrows). Scale bars, 50 μm.

stimulation instead. Conceivably, these issues may be overcome by sophisticated controlled-release devices in the future. Furthermore, in many oral contraceptives progestins are combined with an estrogenic compound, mostly ethinyl estradiol. It is well established that ER and PR interact in breast cancer cells (Giulianelli *et al*, 2012; Mohammed *et al*, 2015; Singhal *et al*, 2018), and such interactions may also operate in the normal breast epithelium as indeed suggested by our finding that an estrogen response was among the gene expression signatures in both mouse and human mammary models. It will be important to explore how these interactions are differentially affected by the different progestins.

Finally, in this study, we mainly assessed the effect of different progestogens on *in vivo* cell proliferation, namely by radiance and Ki67 expression. However, differential effects of the progestins on stem cells, DNA damage, and other biological functions may further contribute to differentially increased breast cancer risk.

Major challenges in working with patient samples are attributable to the large variation between them. For this reason, we started exploring our hypothesis using mammary organoids form inbred mice. As we searched for factors that may determine differential response of different patients' breast epithelial cells, we found no association with age, parity, or race, possibly due to the small size of our cohort. We observed, what remains circumstantial at this point, that xenografts derived from women exposed to specific progestins at the time of surgery responded differently to progestogens, suggesting that individual progestins may induce long-lasting effects in HBECs that affect their response to hormones.

The androgenic progestins DSG, GSN, and LNG induced *Rankl* and *Wnt4* transcript levels in mouse *ex vivo* model, whereas progestins with anti-androgenic properties failed to do so. This finding together with the observation that pharmacological inhibition of AR signaling interferes with *Rankl* induction by R5020 and LNG suggests that full induction of *Rankl* requires active AR signaling

and begs the question how the pure PR agonist is able to induce *Rankl*. It is important to note that we perform *ex vivo* experiments with fresh tissue after short processing. In contrast to well-controlled hormone-free conditions of *in vitro* models and reporter assays, this approach entails carryover of extra- and intracellular testosterone, which likely activates AR signaling. It is self-understood that in the *in vivo* experiments the androgen is always present albeit at low amounts. Together, this suggests that AR signaling needs to be active for PR signaling to occur. At which level (s) the two hormone receptor pathways interact is to be further explored.

When we tested the effects of prolonged *in vivo* exposure to different progestins on HBECs, the 3 testosterone-related compounds, which induced *Rankl* and *Wnt-4* transcripts in the mouse organoids, also increased *in vivo* growth of HBECs. The implication of androgenic activity in tumor promotion is surprising; it is widely assumed that AR antagonizes ER function in the normal breast. Yet our finding may be clinically relevant; the only steroid levels, which correlate with breast cancer risk in women, are those of testosterone (Schernhammer *et al*, 2013). Furthermore, Rankl has been implicated as an important factor early in mammary carcinogensis in mouse models and BRCA1 patients (Gonzalez-Suarez *et al*, 2010; Nolan *et al*, 2016).

More epidemiological studies on the risk associated with distinct hormonal formulations are needed. Our findings are in line with emerging epidemiological evidence pointing to LNG, both in the frame of OC and HRT, as a risk factor for breast cancer with RR of 3.05 (Hunter *et al*, 2010). More specifically, when taking LNG-based medications, ductal breast cancer risk increases to 1.20 and lobular breast cancer risk to 1.33 (Soini *et al*, 2016) and 1.45 (Hunter *et al*, 2010) compared to the general population. Furthermore, our finding that androgenic progestins are more potent than progesterone in increasing HBEC proliferation *in vivo* is in line with epidemiological

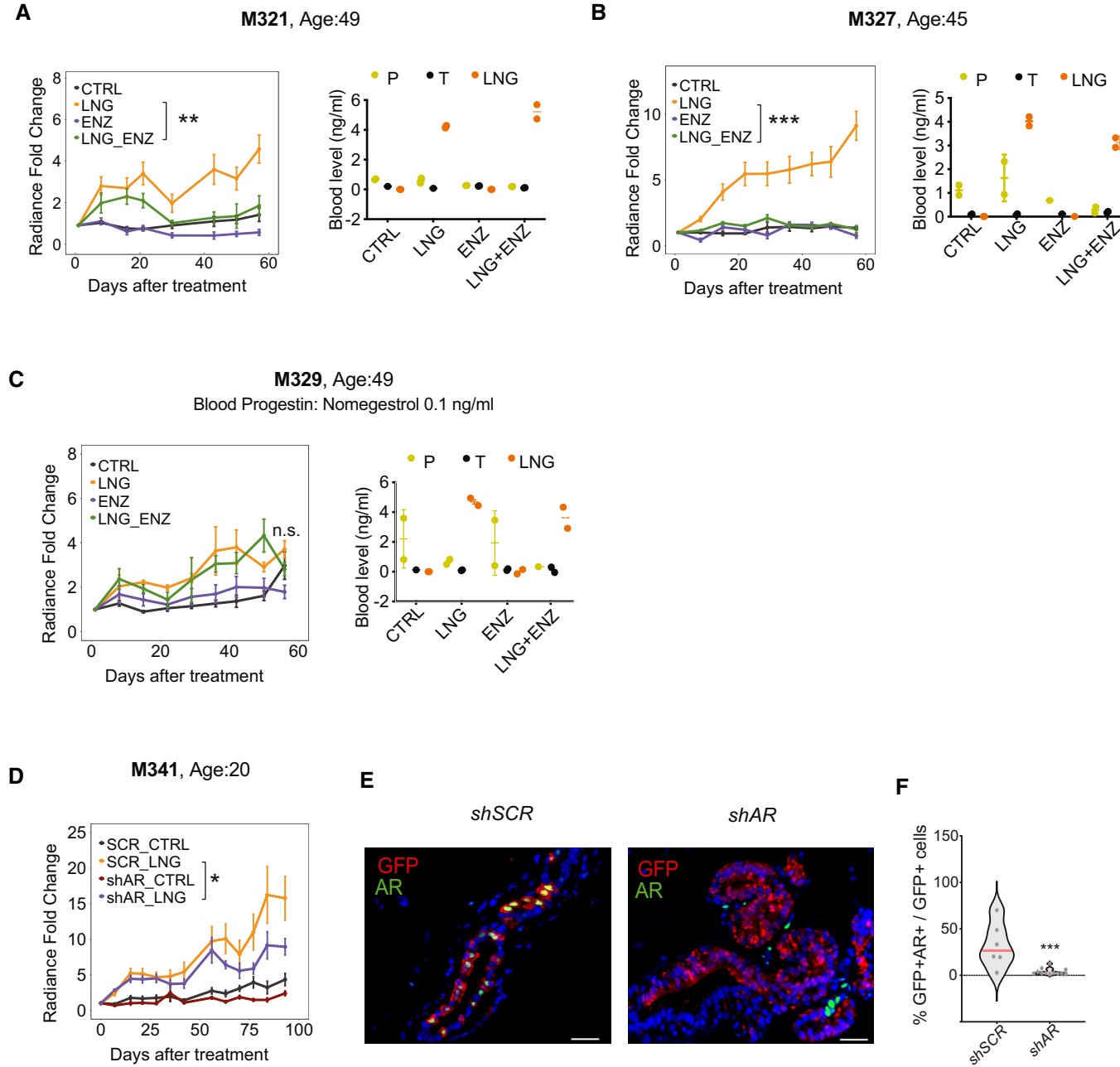

**Figure 7. AR is required for levonorgestrel-driven proliferation of xenografted HBECs.**

A–C    Graphs showing *in vivo* growth of HBECs from different donors as measured by radiance after treatment with LNG, ENZ, or LNG and ENZ, means of radiance in individual glands ± SEM, *n* = 8–10 per treatment, Wilcoxon matched-pairs test. Right: Dot plot showing plasma levels of progesterone (P), testosterone (T) and LNG in individual xenografted animals.

D      Graph showing *in vivo* growth of HBECs from a 20-year-old patient transduced with either *sh scramble* or *shAR* and treated with LNG. Points show means of radiance in individual glands ± SEM; *n* = 8–10 per treatment. Wilcoxon matched-pairs test.

E      Representative micrographs showing co-IF with anti-GFP (red), and anti-AR (green) antibodies on histological sections from glands xenografted with HBECs transduced either with *sh scramble* or *shAR*-expressing lentivirus. Scale bar, 50 μm.

F      Violin plot showing the percentage of AR- and GFP-double+ cells of total GFP+ cells in *sh scramble* (*n* = 6) and *shAR* (*n* = 12) conditions, dots represent individual sectors counted, median (red). Statistical significance was assessed by fitting a generalized linear mixed model with gamma distributions using CTRL as reference. Red line shows median.

Data information: *$P < 0.05$, **$P < 0.01$, ***$P < 0.001$, n.s: not significant.
Source data are available online for this figure.

observations in women on HRT. The French E3N cohort study showed that natural progesterone was not associated with breast cancer risk (Fournier *et al*, 2008), while several studies reported elevated breast cancer risk in women taking androgenic progestins, like LNG and MPA (Rossouw *et al*, 2002). Hence, while caution in extrapolating the experimental findings to the clinics is necessary, our observations are encouraging as they suggest the *ex vivo* and xenograft models presented here are valid and open new opportunities for preventing breast cancer by taking a more informed approach to hormonal contraception.

# Materials and Methods

### Mouse tissue processing and hormone stimulation

Animal experiments were performed in accordance with protocol approved by the Service de la Consommation et des Affaires vétérinaires of Canton de Vaud (VD 1865.3, VD 1865.4). NOD. Cg-Prkdc[scid] Il2rg[tm1Wjl]/SzJ mice (Charles River), BalbC, and C57Bl6 breeders were purchased from Jackson Laboratories. Mammary glands were isolated from 18- to 35-week-old mice, lymph nodes removed, minced with surgical blades, and digested in DMEM/F12, 1% penicillin/streptomycin B with collagenase A 0.25 mg/ml (Roche). Tissue was concomitantly stimulated with hormones for 1, 2, or 6 h and centrifuged at 652 g for 5 min. Fat was removed, and pellet containing organoids washed with phosphate-buffered saline (PBS), resuspended 5 min in 3 ml red cell blood lysis buffer and washed twice with PBS. Hormones and drug used: LNG, Sigma L0551000-30MG; GSN, Sigma SML0292, DSG, Sigma SML0356-5MG; DSP, Sigma SML0147-10MG; CMA, Sigma C5145-1G; CPA, Sigma C3283000-30MG, R5020, Perkin Elmer NLP004005, bicalutamide, Toronto research laboratory B382000, enzalutamide, Selleck Chemicals S1250.

### Patient sample processing

All experiments conformed to the principles set out in the WMA Declaration of Helsinki and the Department of Health and Human Services Belmont Report. The cantonal ethics committee approved the study (183/10). Breast tissue was obtained from women undergoing reduction mammoplasties with no previous history of breast cancer, who gave informed consent. Samples were examined by the pathologist to be free of malignancy and processed as described (Sflomos *et al*, 2015). Human tissue samples were mechanically and enzymatically dissociated, single cells were prepared, counted, and spin infected using Lenti-ONE CMV-GFP(2A)Luc2 WPRE VSV, V621004001, or pR980 Luc2GFP (GEGTECH) at a multiplicity of infection of 1 at 2.500 rpm for 3 h at room temperature in 500 µl of serum-free medium. Following overnight incubation at 37° C, cells were resuspended in PBS to final concentration of $4–5 \times 10^5$ viable cells per 10 µl. Hormone stimulations and isolation of EPCAM[+] breast cells were performed as described (Tanos *et al*, 2013).

### Intraductal xenografts

Mice were anesthetized by intraperitoneal injection with 10 mg/kg xylazine and 90 mg/kg ketamine (Graeub) and injected intraductally with 10 µl cell suspension per gland. Bioluminescence imaging was

performed as described (Sflomos *et al*, 2016). After 3–10 weeks, when bioluminescence from individual glands reached $10^7$ total flux (p/s/cm2/sr), animals were assigned to different treatment groups and hormone pellets implanted subcutaneously. Enzalutamide (Selleckchem, Catalog No. S1250) dissolved in 15% DMSO in PEG300 was injected intraperitoneally 3 times/week at 30 mg/kg. The long-term LNG-stimulation was started 4 months after intraductal injection and conducted over 6 months with 3 subsequent pellet implants.

### Hormone pellets

Pellets were prepared by mixing the required amount of part A, MP3745/E81949 and part B, MP3744/E81950 of low consistency silicon elastomer (MED-4011) and hormone powder to homogeneity. Mix was transferred to a syringe and incubated at 37°C overnight. The silicon-hormone mass was removed from the syringe and cut to pieces, as described (Duss *et al*, 2007). The original protocol was adapted for different doses as described below.

| Hormone pellet | Catalog number | Hormone dose/ pellet (mg) | Silicon part A (mg) | Silicon part B (µl) | Hormone powder (mg) | Weight (mg) |
|---|---|---|---|---|---|---|
| LNG | 1362602 | 9 | 223 | 23.75 | 200.92 | 23.5 |
| GSN | Y0001112 | 4.5 | 223 | 23.75 | 85.47 | 18.25 |
| DSG | Y0000509 | 9 | 223 | 23.75 | 171 | 26.10 |
| CMA | C343500 | 0.5 | 587.5 | 62.5 | 157.36 | 3 |
| CPA | C989100 | 0.6 | 587.5 | 62.5 | 117 | 4.6 |
| DSP | D689500 | 20 | 587.5 | 62.5 | 587.5 | 49.50 |
| Progesterone | P0130-25G | 20 | 3525 | 375 | 3525 | 49.5 |

### Mammary gland whole mounts

Mammary gland whole mounts were performed as described previously (Ayyanan *et al*, 2011); stereomicrographs were acquired on a LEICA MZ FLIII stereomicroscope with Leica MC170 HD camera. Fluorescence stereomicrographs were acquired using a LEICA M205FA with a Leica DFC 340FX camera.

### Immunohistochemical staining

Tissues were fixed for 2 h at room temperature in 4% paraformaldehyde and paraffin-embedded. Dewaxed and rehydrated paraffin sections were pretreated in 10 mM Na citrate (pH 6.0) at 95° for 20 min. Blocking was performed with 1% BSA for 60 min. Primary antibodies used were as follows: Ki67 (rabbit α-Ki67, clone Sp6, Spring Amsbio, dilution 1:200), ER (rabbit α-ER, clone Sp1, Zytomed RTU), PR (rabbit α-PR, clone 1E2, Ventana RTU), AR (rabbit α-AR, clone SP107, Thermo Fisher, diluted 1:200), CK7 (rabbit α-CK7, clone Sp52, Abcam, diluted 1:500), CK19 (mouse KS19.2 (Z105.6), American Research Products Inc. (diluted 1:50), and mouse α-ECAD (clone G-10, sc-8426, diluted 1:100). For IHC, chromogenic revelation was performed with ChromoMap DAB kit (Roche Diagnostics, Switzerland) and sections were counterstained with Mayer's hematoxylin. For fluorescence microscopy, nuclei were counterstained with DAPI (Sigma). IHC for ER, PR, and AR was performed on Discovery Ventana ULTRA (44). IHC images were acquired on Leica DM 2000 with Leica DFC450 C camera. IF images

were acquired on Zeiss LSM700 confocal microscope. Slides were scanned with Olympus VS120-L100 slide scanner with 20×/0.75 objective and Pike F505 C Color camera. Images were loaded into QuPath using the BioFormats extension (Bankhead *et al*, 2017).

**Hormone measurements**

Plasma steroid concentrations were measured by LC–MS High Resolution (Q-Executive, Thermo Fisher Scientific) for testosterone, progesterone, and 17β-estradiol as described in (Cagnet *et al*, 2018) and for progestins as described in (Laszlo *et al*, 2019).

**RNA extraction, cDNA preparation, and RT-PCR**

Human breast microstructures and mouse organoids were homogenized in Trizol (Invitrogen). Aqueous phase containing RNA was chloroform extracted and processed with miRNeasy extraction kit (Qiagen). cDNA was synthesized using random p(dN)6 primers (Roche) and MMLV reverse transcriptase (Invitrogen). SYBR Green PCR Core Reagent System (Qiagen) was used for semi-quantitative real time (RT-PCR) using the following primers.

| Gene | Forward primer | Reverse primer |
|---|---|---|
| *M 36B4* | GTG TGT CTG CAG ATC GGG TA | CAG ATG GAT CAG CCA GGA AG |
| *M RANKL* | CCC ACA ATG TGT TGC AGT TC | TGT ACT TTC GAG CGC AGA TG |
| *M WNT4* | AGG AGT GCC AAT ACC AGT TCC | CAG TTC TCC ACT GCT GCA TG |
| *HPRT* | GAC CAG TCA ACA GGG GAC AT | CCT GAC CAA GGA AAG CAA AG |
| *GAPDH* | CCC CAC TTG ATT TTG GAG GGA | AGG GCT GCT TTT AAC TCT GGT |
| *CSN3* | AAC AAC CAG CAT GCC ATG AG | AAC AAC CAG CAT GCC ATG AG |
| *CSN1S1* | GAG GCT TCT CAT TCT CAC CTG TC | ACT GCT CTC TGA TGG ATT CTG AAG |
| *LALBA* | AGG TCC CTC AGT CAA GGA ACA | GGC TTT ATG GGC CAA CCA GT |

**RNA *in situ* hybridization**

RNAscope assay (Advanced Cell Diagnostics, Cat. No. 323110) was performed according to manufacturer's protocol on 4 μm deparaffinized sections and with probes for *RANKL* (ACD, Cat No. 523331-C2), *WNT4* (ACD, Cat No. 429441), Mm-Ppib (ACD, Cat. No. 313911, positive control), and DapB (ACD, Cat. No. 310043, negative control) at 40°C for 2 h and revealed with TSA Plus-Cy3 (Perkin Elmer, Cat. No. NEL744001KT). *WNT4* and *RANKL* were revealed with Opal570 and Opal650, respectively. Confocal images were acquired using a laser-scanning confocal inverted microscope (LSM700, Carl Zeiss, Inc.).

**RNAseq experiment**

RNAseq libraries were prepared using Truseq Stranded RNA producing single-end reads of 100 bp and sequenced on Illumina Hiseq 2500 instrument. RNA was extracted from the epithelial-enriched mouse organoids using miRNeasy extraction kit (Qiagen) and from EPCAM+ cells using miRNeasy Mini Kit (Qiagen).

**Computational analysis**

Sequencing reads were quality controlled using FastQC (v0.11.8) (Wingett & Andrews, 2018). Mouse raw reads were aligned to the mouse genome (mm9) and human raw reads to the human genome (hg19) using TopHat (v2.0.11) (Kim *et al*, 2013). Gene counts were generated using FeatureCounts (v1.5.1) (Liao *et al*, 2014). Normalization for sequencing depth differences and differential expression analysis was performed using the DESEQ2 (v3.11) package from Bioconductor (Love *et al*, 2014). Genes were considered differentially expressed based on adj.*P*.val < 0.05. GSEA was carried out using ClusterProfiler. Default parameters were applied (Yu *et al*, 2012). For GO enrichment analysis, hallmark gene sets and the C2 curated gene set collections from the MSigDB v6.2 were used. Heatmaps were generated using Complex Heatmap package in R (Gu *et al*, 2016), and the normalized counts were illustrated upon conversion to Z-Score according to row values. Euclidean distance was used to perform unsupervised hierarchical clustering. For fitting regression models, the lm function from the stats package in R was used.

## Data availability

Gene Expression Omnibus (GEO) accession number for the transcriptomics data reported in this study is GSE155274 (http://www.ncbi.nlm.nih.gov/geo/query/acc.cgi?acc=GSE155274); GEO SubSeries accession numbers referring to the mouse organoids and human

**The paper explained**

**Problem**

Breast cancer is the most commonly diagnosed cancer worldwide. Hormonal contraception, which is used by a growing number of women worldwide, increases relative breast cancer risk of current users by 20-30%. Hormonal contraceptives contain one of several progestins, compounds with activities of the endogenous hormone progesterone that are structurally different and may hence differentially affect breast cancer risk. We have lacked models to study which substances exert protective and which tumor-promoting effects in the breast epithelium.

**Results**

We have used a combination of mouse mammary gland tissue *ex vivo* assays and developed human breast epithelial implants into mice to test the ability of 6 widely used progestins to induce expression of functionally important progesterone receptor target genes and to stimulate breast epithelial cell proliferation over prolonged periods of time. We find that progestins routinely included in contraceptives can be sub-divided in two categories: 1) those with activities overlapping with testosterone, androgenic progestins, which induce expression of important mediators of progesterone receptor signaling and consistently elicit proliferation of human breast epithelial cells; 2) those with activities blocking testosterone, anti-androgenic progestins, which do not have these effects.

**Impact**

Our finding that different progestins have distinct biological activities in the human breast epithelium depending on their androgenic properties suggests that breast cancer associated with hormonal contraception can be prevented by making more informed choices and avoiding androgenic compounds. The *in vivo* xenograft models we present open unprecedented possibilities to analyze the biological actions in the breast epithelium of any substance women are exposed to and help gain insights into how to prevent breast cancer.

microstructures experiments are GSE155272 (http://www.ncbi.nlm. nih.gov/geo/query/acc.cgi?acc=GSE155272) and GSE155273 (http:// www.ncbi.nlm.nih.gov/geo/query/acc.cgi?acc=GSE155273) .

**Expanded View** for this article is available online.

## Acknowledgments

We thank B. Mangeat at the EPFL gene expression core facility, J. Dessi-moz at the EPFL histology core facility, and H. Henry, R. Nellen for technical assistance. M. S. and V.S. were supported by SNF 31003A_141248 Hormonal and cell signaling control of mammary gland morphogenesis: ER/PR and Notch signaling interactions and by Swiss Cancer League KFS-3701-08-2015: Lobular carcinoma of the breast: insights from a new PDX model, F.DM., C. L., and C.C. by SNF 310030_179163/1: Exploring key steps of the metastatic cascade in ER+ breast cancer in vivo, L.B by KFS-4738-02-2019-R: Different facets of ER signaling during ER+ breast carcinogenesis, G.S and V. S. by Biltema ISREC Foundation Cancera Stiftelsen, Mats Paulssons Stiftelse, and Stitelsen Stefan Paulssons Cancerfond, and A-S. L by scholarship JALI.

## Author contributions

Conceptualization MS, FDM., VS, CC, CB; Formal Analysis MS, GA, PB, FDM; Investigation MS, FDM, CL, CC, A-SL, VS, AA, GS, MF; Resources LB, WR, M-CG-M, MF, Writing MS, FDM, CB; Funding Acquisition CB.

## Conflict of interest

The authors declare that they have no conflict of interest.

## For more information

https://www.europadonna.org/

Gene list:
*RANKL/TNFSF11* - https://www.ncbi.nlm.nih.gov/gene/8600
*Tnfsf11* - https://www.ncbi.nlm.nih.gov/gene/21943
*WNT4* - https://www.ncbi.nlm.nih.gov/gene/54361
*Wnt4* - https://www.ncbi.nlm.nih.gov/gene/22417
*ESR1* - https://www.ncbi.nlm.nih.gov/gene/2099
*PGR* - https://www.ncbi.nlm.nih.gov/gene/5241
*KI67* - https://www.ncbi.nlm.nih.gov/gene/4288
Author's websites:
https://www.epfl.ch/labs/brisken-lab/
https://www.epfl.ch/labs/brisken-lab/preclinicalmodelcourse/
https://www.cancerprev.com/
https://cancerprev.ch/

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
