## [Review Process File · EMBO Molecular Medicine]

Contraceptive progestins with androgenic properties stimulate breast epithelial cell proliferation

Marie Shamseddin, Fabio De Martino, Céline Constantin, Valentina Scabia, Anne-Sophie Lancelot, Csaba Laszlo, Ayyakkannu Ayyanan, Laura Battista, Wassim Raffoul, Marie-Christine Gailloud-Matthieu, Philipp Bucher, Maryse Fiche, Giovanna Ambrosini, George Sflomos, and Cathrin Brisken

DOI: [10.15252/emmm.202114314](https://doi.org/10.15252/emmm.202114314)

Corresponding author(s): Cathrin Brisken (cathrin.brisken@epfl.ch)

Review Timeline:

Submission Date:	23rd Mar 21
Editorial Decision:	24th Mar 21
Revision Received:	5th Apr 21
Editorial Decision:	9th Apr 21
Revision Received:	15th Apr 21
Accepted:	23rd Apr 21

Editor: Jingyi Hou

Transaction Report:

(Note: Please note that the manuscript was previously reviewed at another journal and the reports were taken into account in the decision making process at EMBO Molecular Medicine. Since the original reviews are not subject to EMBO's transparent review process policy, the reports and author response cannot be published. With the exception of the correction of typographical or spelling errors that could be a source of ambiguity, letters and reports are not edited. Depending on transfer agreements, referee reports obtained elsewhere may or may not be included in this compilation. Referee reports are anonymous unless the Referee chooses to sign their reports.)

1st Editorial Decision**24th Mar 2021**

Thank you for submitting your manuscript to EMBO Molecular Medicine. I have now carefully read your manuscript and the response to the reviewers, and discussed them with the other members of our editorial team. In addition, I have also consulted a member of our Editorial Advisory Board. I am pleased to inform you that we find your manuscript suitable for publication in EMBO Molecular Medicine.

1st Authors' Response to Reviewers**5th Apr 2021**

The authors have made all requested editorial changes.

1st Revision - Editorial Decision**9th Apr 2021**

Thank you for the submission of your revised manuscript to EMBO Molecular Medicine. I am pleased to inform you that we will be able to accept your manuscript pending the following editorial level amendments:

2nd Authors' Response to Reviewers**15th Apr 2021**

The authors have made all requested editorial changes.

Accepted**23rd Apr 2021**

23rd Apr 2021

I am pleased to inform you that your manuscript is accepted for publication and is now being sent to our publisher to be included in the next available issue of EMBO Molecular Medicine.

YOU MUST COMPLETE ALL CELLS WITH A PINK BACKGROUND ↓
PLEASE NOTE THAT THIS CHECKLIST WILL BE PUBLISHED ALONGSIDE YOUR PAPER

Corresponding Author Name: Cathrin Brisken, MD, PhD
Journal Submitted to: EMBO Molecular Medicine
Manuscript Number: EMM-2021-14314